# Stray Dogs as Reservoirs and Sources of Infectious and Parasitic Diseases in the Environment of the City of Uralsk in Western Kazakhstan

**DOI:** 10.3390/biology14060683

**Published:** 2025-06-11

**Authors:** Askar Nametov, Rashid Karmaliyev, Bekzhassar Sidikhov, Kenzhebek Murzabayev, Kanat Orynkhanov, Bakytkanym Kadraliyeva, Balaussa Yertleuova, Dosmukan Gabdullin, Zulkyya Abilova, Laura Dushayeva

**Affiliations:** 1Institute of Veterinary and Agrotechnology, Zhangir Khan West Kazakhstan Agrarian Technical University, 51 Zhangir Khan Street, Uralsk 090009, Kazakhstan; anametov@mail.ru (A.N.); sidihovbm@mail.ru (B.S.); murzabaev.k@mail.ru (K.M.); bkadralieva@mail.ru (B.K.); aliba.87@mail.ru (B.Y.); dosya_gabdullin@mail.ru (D.G.); 2Veterinary and Zooengineering Faculty, Kazakh National Agrarian Research University, 8 Abay Avenue, Almaty 050010, Kazakhstan; k_orynkhanov@mail.ru; 3Departament of Veterinary Medicine, Faculty of Agricultural Science, Kostanay Regional University Named After Akhmet Baitursynuly, 10 Chkalova Street, Apartment 67, Kostanay 110005, Kazakhstan; dgip2005@mail.ru

**Keywords:** stray dogs, zoonotic diseases, PCR, ELISA, helminthiasis, *B. canis*, *T. canis*, *E. granulosus*, epizootiology

## Abstract

In many large cities, the population of stray dogs is rapidly increasing. This rise poses challenges not only to the environment and public safety but also to public health. Stray dogs often carry diseases that can be transmitted to humans and other animals. This study investigated the role of stray dogs in the spread of dangerous infections in the city of Uralsk, Kazakhstan. Over one thousand stray dogs were captured, and biological samples were tested to determine the presence of harmful bacteria and parasites. The results indicated that many dogs exhibited signs of serious diseases, including zoonotic infections such as brucellosis and leptospirosis. Additionally, some dogs were found to harbor parasitic worms capable of causing illness in humans. These findings demonstrate that stray dogs can act as reservoirs for diseases and potentially pose a health risk to urban populations. Consequently, the effective management and control of stray dog populations are essential to enhance public health protection. This study underscores the importance of monitoring street animals and implementing preventive measures to reduce the transmission of diseases to humans.

## 1. Introduction

The increasing number of domestic and stray dogs in large cities has become an increasingly pressing issue. This phenomenon is linked to rising population densities, the characteristics of urban environments, and insufficient control over animal reproduction. In addition to social and ecological aspects, this trend has significant epidemiological implications, as dogs can act as carriers of various infectious and parasitic diseases [1].

Cities with high population densities create favorable conditions for the transmission of zoonotic infections between dogs and humans. Unauthorized dumps, lack of veterinary control, and overcrowding of animals contribute to the spread of viruses, bacteria, and parasites. Dogs, especially stray ones, become reservoirs for pathogens that can be transmitted to humans through bites, contact with fur, or contaminated excretions. Deteriorating sanitary conditions and lack of vaccination increase the risks of outbreaks. Studying the role of dogs in the spread of infections is crucial for developing effective preventive measures. Dog population control and mandatory vaccination can help reduce the risks of diseases among both humans and animals. The rising number of dogs in large cities requires a comprehensive approach, including sanitary control measures, epidemiological monitoring, and animal population regulation. Only through the combination of these strategies can we minimize the threat of zoonotic infections and create a safe environment for both people and animals [2].

A study of dogs and cats in Italy found infections in domestic animals with zoonotic parasites, including *Toxocara canis* (Werner, 1782), *Toxocara cati* (Schrank, 1788), *Trichuris vulpis* (Froelich, 1789), *Ancylostoma caninum* (Ercolani, 1859), and *Giardia duodenalis* (Stiles, 1902) [3]. In West Africa, specifically in Nigeria, stray dogs were found to be infected with *A. caninum*, with an infection rate of 62.5%, *T. canis*, at 20.8%, *Dipylidium caninum* (Linnaeus,1758), at 18.7%, and *Strongyloides stercoralis* (Bavay, 1876), at 2.0% [4]. A study conducted in the capital of Argentina, covering 219 dogs from socioeconomically disadvantaged areas of Buenos Aires with a high level of unmet basic needs, revealed the presence of antibodies to *Brucella canis* (Carmichael and Bruner, 1968) in 7.3% of the animals. Additionally, in three cases, *B. canis* was found to be excreted in bacterial form. These findings highlight a potential epidemiological threat to the population at risk of infection [5,6,7].

In Canada, cases of human infection with *Echinococcus multilocularis* (Leuckart, 1863), have been reported, with transmission occurring via infected dogs. A study was conducted to assess the prevalence of helminth infection and related risk factors among domestic dogs in Calgary, Alberta. The study, which involved collecting and analyzing fecal samples from dogs while considering potential risk factors, found that 13 out of 696 samples collected in August and September 2012 tested positive for *E. multilocularis* by PCR analysis, with a helminth infection rate ranging from 2.4% to 95% [8,9].

In the city of Novosibirsk (Russia), an assessment of the prevalence of *Opisthorchis felineus* (Rivolta, 1884) infection was carried out among 103 cats and 101 dogs taken from shelters in various city districts as well as rural settlements along the Ob River. It was found that the infection rate of *O. felineus* in cats ranged from 12.6% to 95%, significantly higher than that in dogs, where the infection rate ranged from 4.0% to 95% [10].

In Western Kazakhstan, foci of opisthorchiasis are most prevalent among the population and carnivorous animals in river basins that provide favorable conditions for mollusks and carp species. It has been established that in the coastal villages along the Ural River, the prevalence of *O. felineus* infection in dogs was, on average, 89.7%, with an intensity of 19.6 individuals per dog. In cats, the prevalence was, on average, 97.9%, with an intensity of 34.4 individuals per cat [11]. The highest prevalence of echinococcosis has been recorded in the southern regions of Kazakhstan. An analysis of zoonotic helminthiasis cases from January to May 2015 found that echinococcosis was diagnosed in 65 individuals across the country. Among these, 42 cases (64.6%) were registered among rural residents, who have more direct contact with the primary carriers of the infection—dogs. In the structure of the affected population, 14 children (21.5%) under the age of 14 and 5 adolescents (7.7%) aged 15–17 were identified [12]. The epizootiological situation in the Western Kazakhstan region regarding parasitic and infectious diseases in dogs within the urban environment remains tense. According to various studies, several species of helminths from the classes Trematoda (Rudolphi, 1808), Cestoda (Cobbold, 1864), and Nematoda (Rudolphi, 1808) have been identified in dogs, which continues to be a significant concern [13,14,15,16].

The first case of the molecular identification of *Dirofilaria repens* in the heart of a wild wolf in Kazakhstan indicates an expansion of the parasite’s range and underscores the epizootiological significance of this finding. Morphological identification proved insufficient, and only molecular methods based on the SSU rRNA gene enabled precise species determination. These data confirm the necessity for the widespread application of molecular technologies in the diagnosis of dirofilariasis and call for further investigation of the infection’s distribution in the region [17]. C.H. Lai et al. (2001) reported a significant increase in the number of dogs infected with Dirofilaria in Taiwan over recent years [18]. I. Tsai et al. (2003) conducted necropsies on 82 stray dogs in the vicinity of Taichung, Taiwan [19]. Dirofilaria parasites were found in the pulmonary artery or right atrium in 54% of the infected dogs. Blood smear examinations detected microfilariae of *Dirofilaria immitis* (Leidy, 1856) in 63.6% of cases. According to C.C. Wu and P.C. Fan (2003), the dirofilariasis prevalence in dogs across Taiwan ranges from 4 to 41%, with a tendency toward further spread [20]. In Kalmykia (Russia), studies by I.A. Arkhipov, V.A. Bashankaev, D.R. Arkhipova (2002), and D.R. Arkhipova (2003) established that dogs of all age groups, except those under one year old, are infected with *D. repens* (Railliet and Henry, 1911) [21]. The prevalence of dirofilariasis during helminthological necropsies was 33.3%, with an intensity of infection averaging 15.3 ± 2.6 specimens per animal [22,23,24,25].

In Western Kazakhstan, *D. repens* was detected in stray dogs during helminthological necropsies, with an infection prevalence of 29.4% [16]. The widespread occurrence of parasitic and infectious diseases among dogs is driven by the growing population of stray animals. These animals act as reservoirs for pathogens of infectious and parasitic diseases, and their uncontrolled movement throughout the city contributes to the further spread of these pathogens. An additional factor exacerbating the epizootiological situation is the insufficient level of veterinary control, as well as the poor conditions for the care of domestic (yard) dogs and the complete absence of such conditions in stray animals [26].

In light of this, we initiated our own research in 2024 aimed at investigating and verifying the reliability of data on the prevalence of infectious and parasitic diseases among stray dogs in the city of Uralsk.

## 2. Materials and Methods

This study was conducted in 2024 at the veterinary clinic and laboratory of the Testing Center at Zhangir Khan West Kazakhstan Agrarian Technical University (Zhangir Khan University), located in the city of Uralsk, Kazakhstan.

### 2.1. Sample Collection and Clinical Examination

The capture of stray dogs in the city of Uralsk, Republic of Kazakhstan, was carried out in accordance with the “On Responsible Treatment of Animals” No. 97-VII Law of the Republic of Kazakhstan (LRK), dated 30 December 2021 [27]. Stray dogs were defined as animals found in public spaces without supervision by an owner responsible for their behavior and for preventing potential conflicts with humans. These activities were conducted under approved population control programs, with mandatory coordination with local executive authorities and prior ecological assessments of target areas.

Only humane capture techniques were employed, strictly designed to avoid injury or stress to the animals. These included the use of soft nets and loops, baited trap cages, and tranquilizers—administered exclusively under the supervision of a licensed veterinary professional. A veterinarian was present at all stages of the capture and transportation to monitor the animals’ health and welfare. Capture procedures were conducted primarily in the early morning or evening hours to minimize stress factors such as loud noises or physical force.

Captured animals were transported to temporary holding facilities compliant with veterinary and sanitary regulations. The core approach followed the TNR (Trap–Neuter–Return) strategy, under which animals underwent clinical examination, sterilization, vaccination (including rabies), and microchipping and were returned to their original habitat, provided they showed no signs of aggression or disease. These measures were aligned with national legislation and international principles of humane animal treatment, ensuring rational population control and the prevention of epizootic and public health risks.

Capture sites were selected based on known dog habitat zones, as registered with the veterinary service of Uralsk (Figure 1). All captured dogs underwent clinical examination; those showing signs of illness were quarantined, while clinically healthy individuals were selected for further diagnostic testing within the scope of this scientific investigation.

The clinical examination was conducted following established veterinary standards. The general condition of the dogs was assessed, visible mucous membranes were inspected (oral, nasal, conjunctiva of the eyes, and for females—the vagina), the body temperature was measured, and pulse and respiratory rates were recorded. All parameters were entered into an electronic database. Over the course of one year, 1213 stray dogs were captured from various districts of the Uralsk metropolitan area. Biological samples (blood, urine, and feces) were collected from the captured animals to identify infectious and parasitic pathogens using molecular and helminthological diagnostic methods (see Table 1).

#### 2.1.1. Blood Collection

Before blood sampling, dogs were gently restrained to minimize stress and prevent injury to both the animal and personnel. Venipuncture was performed under sterile conditions from the subcutaneous vein of the dog’s forepaw. Whole blood was collected into sterile plastic vacuum tubes (vacutainers) containing the anticoagulant ethylenediaminetetraacetic acid (EDTA, K_2_ salt), with a volume of 5 mL and a purple cap. After collection, the tubes were gently inverted 5–8 times to prevent clotting. Blood samples were stored in a refrigerator at +2 to +8 °C to prevent hemolysis and the alteration of the blood cell composition. To obtain serum, blood was drawn using the above method into VACUETTE tubes without anticoagulants and subsequently centrifuged or 10 min at 3000 rpm. The separated serum was transferred into sterile tubes using an automatic pipette and stored at −20 °C until analysis [28].

#### 2.1.2. Urine Collection

Urine was collected by catheterization of the urethra. Samples were transferred into sterile containers and stored at +4 °C until further analysis. For male dogs, after restraining the animal in a standing position, the preputial area was disinfected with an antiseptic solution. The catheter was lubricated with sterile lubricant and gently inserted into the urethra, advancing toward the bladder. Correct placement in the bladder was confirmed by the appearance of urine in the catheter. For female dogs, restraint was performed in lateral recumbency. The external genital area was disinfected. The urethral opening was identified on the ventral wall of the vagina approximately 1–2 cm from the vaginal entrance, using a vaginal speculum if necessary. The catheter was carefully inserted until urine appeared, confirming successful entry into the bladder [29].

### 2.2. Coprological Pathogen Detection

Fecal samples were collected rectally using a sterile cotton swab inserted 2–3 cm into the rectum and then gently withdrawn and placed into a sterile container for further analysis. Samples were stored at −20 °C. All samples were labeled with the dog’s identification number and the date of collection. 

Detection of Helminth Eggs in Dog Feces by Fülleborn’s Flotation Method Using a Helminth Egg-Counting Chamber.

The detection of helminth eggs in dog feces was carried out using the Fülleborn flotation method with a specialized helminth egg-counting chamber. A 3 g sample of feces was placed in a mortar and mixed with a small volume (5 mL) of a flotation solution composed of sodium chloride (specific gravity: 1.25) and ammonium nitrate (specific gravity: 1.38). The sample was thoroughly homogenized using a pestle. While stirring, an additional flotation solution was gradually added. The resulting suspension was filtered through a metal sieve into a 30 mL beaker and left to settle. After 5–10 min, 3–5 drops were taken from the surface of the suspension using a metal loop (one from the center and the rest from the periphery). The drops were placed into one of the compartments of the lower plate of the helminth egg-counting chamber. The chamber was then covered with the upper plate, and the additional flotation solution was gently added with a pipette until surface tension was established. After approximately one minute, helminth eggs floated to the underside of the top plate.

The chamber was then examined under a microscope, and all eggs within the compartment were counted. The total number of eggs was divided by the number of drops placed in the chamber, and the result was multiplied by a conversion factor of 38 (the approximate number of loopfuls that could be placed on the surface of the suspension in the beaker). The final result represented the number of eggs per 3 g of feces.

The metal loop, mortar, pestle, and sieve were thoroughly washed before processing each sample.

In total, 102 fecal samples from stray dogs were examined. Helminth infection was determined using the Fülleborn flotation technique—one of the standard methods for detecting helminth eggs.

Two key indicators were used in the analysis:

EI (extensiveness of infection, %): the proportion of infected animals;

II (intensity of infection, eggs/animal): the average number of parasite eggs per infected individual.

Based on their life cycles, the helminths were classified into the following:

Helminths with an indirect life cycle (involving intermediate hosts): *O*. *felineus*, Taeniidae, and *D*. *caninum*;

Soil-transmitted helminths (STHs) or helminths with a direct life cycle: *T*. *leonina*, *T*. *canis*, and *A*. *caninum* [30,31].

### 2.3. Serological Pathogen Detection

Detection of Infectious Disease Agents in Dog Serum Using Enzyme-Linked Immunosorbent Assay (ELISA). 

The presence of antibodies to various infectious agents—*Pasteurella multocida*, *Leptospira* spp., *Chlamydia trachomatis*, *Brucella* spp., *Listeria monocytogenes*, and *Mycobacterium* spp.—was investigated in canine serum samples using the enzyme-linked immunosorbent assay (ELISA). A total of 102 serum samples from dogs were examined for antibodies against these pathogens. 

ELISA Procedure for the Detection of Antibodies in Dog Serum. 

To identify pathogen-specific antibodies, a commercial diagnostic ELISA kit (MVA group, Almaty, Kazakhstan) was used: “Diagnostic kit for the detection of individual specific IgG antibodies to bacteria in serum (plasma) of carnivores (dogs and cats) by enzyme immunoassay (ELISA)”. The assay was performed according to the manufacturer’s instructions as follows: All necessary consumables and reagents were first prepared, including the diagnostic kit and a microplate pre-coated with bacterial antigens. The serum samples collected from stray dogs were clarified by centrifugation to remove cellular debris. Each well of the microplate was loaded with 100 μL of serum and incubated at 37 °C for 30 min to allow antibodies in the sample to bind to the immobilized antigens. After incubation, the plate was washed several times to remove unbound materials. Then, enzyme-conjugated secondary antibodies (anti-IgG) were added. A second incubation at 37 °C for 30 min was performed. Following a subsequent washing step, a chromogenic substrate (TMB) was added. Antibody complexes of the substrate underwent a colorimetric change. The substrate incubation was carried out at room temperature for 15–30 min. The enzymatic reaction was then stopped with a stop solution. The optical density (OD) was measured at 450 nm using a microplate reader (“Sib-Photometer”) manufactured by LLC NPF “Sibbiotest”, Novosibirsk, Russia. This reader is designed for ELISAs using 96-well plates and supports with OD measurements at 405, 450, 492, and 630 nm. It provides rapid readings, processing one plate in approximately 5 s. Positive and negative control samples were included in the assay to ensure accuracy and reliability. All procedures strictly followed the kit manufacturer’s protocol to prevent experimental errors and ensure the validity of the results [32].

Detection of Parasitic Infections (*E*. *granulosus* and *T*. *canis*) in Dog Serum Using Enzyme-Linked Immunosorbent Assay (ELISA).

The detection of antibodies against the parasitic pathogens *Echinococcus granulosus* and *T*. *canis* in dog serum was carried out using the enzyme-linked immunosorbent assay (ELISA), following the procedure described above. The serological testing aimed to assess the level of seropositivity to these specific parasitic infections in the sampled dog population.

A total of 102 serum samples from stray dogs were analyzed. Two parasites—*T. canis* and *E. granulosus*—were selected as the target organisms for this investigation. The ELISA was performed according to the manufacturer’s protocol, using commercially available diagnostic kits designed to detect specific IgG antibodies against these helminths in canine serum. The test procedure included antigen-coated microplates, incubation with dog serum, the addition of enzyme-conjugated secondary antibodies, substrate reaction, and measurement of the optical density at 450 nm using a microplate reader.

The results provided data on the seroprevalence of *T*. *canis* and *E*. *granulosus*, indicating the exposure and potential risk of transmission in the studied canine population.

### 2.4. Molecular Pathogen Detection

Detection of Infectious Agents (*P*. *multocida*, *Leptospira* spp., *C*. *trachomatis*, and *Brucella* spp.) in Dog Blood and Urine Samples by PCR. 

DNA extraction from blood and urine samples for the detection of parasitic pathogens was performed using the QIAamp DNA Mini Kit (Qiagen, Hilden, Germany) following the manufacturer’s instructions.

PCR Amplification of the 16S rRNA Gene for the Detection of Bacterial Pathogens.

A broad-range PCR targeting the 16S rRNA gene was used for the initial screening of bacterial infections. The amplification was carried out using specific oligonucleotide primers: forward primer 16S rRNA-F and reverse primer 16S rRNA-R, producing a 1500 bp amplicon. The PCR was performed with the ScreenMix kit (Evrogen, Moscow, Russia). The reaction mixture (25 µL total volume) consisted of 5 µL of 5× mix, 0.5 µL of each primer, 3 µL of DNA template, and 16 µL of nuclease-free water.

The PCR thermal cycling conditions were as follows:

Initial denaturation at 95 °C for 3 min;

A total of 30 cycles of denaturation at 96 °C for 15 s, annealing at 58 °C for 20 s, and elongation at 72 °C for 90 s;

Final extension at 72 °C for 1 min.

Amplicons were stored at 4 °C until further analysis [33].

PCR Detection of *P*. *multocida*.

For the specific detection of *P*. *multocida*, the primers KMT1T7 (forward: ATC CGC TAT TTA CCC AGT GG) and KMT1SP6 (reverse: GCT GTA AAC GAA CTC GC AC) were used, targeting a 456 bp fragment.

The PCR was carried out in a total volume of 25 µL, containing 2.5 µL of 10× buffer, 1 µL of each primer, 0.2 µL of Assu Prime Taq polymerase, 3 µL of DNA, and 17.3 µL of nuclease-free water. The amplification conditions were as follows:

Initial denaturation at 94 °C for 5 min;

A total of 40 cycles of denaturation at 94 °C for 30 s, annealing at 55 °C for 30 s, and elongation at 68 °C for 1 min;

Final extension at 68 °C for 10 min [34,35].

PCR Detection of *Leptospira* spp.

The primers G1 (forward: CTG AAT CGC TGT ATA AAA GT) and G2 (reverse: GGA AAA CAA ATG GTC GGA AG) were used to amplify a 285 bp product specific for *Leptospira* spp.

The reaction mixture included 5 µL of 5× ScreenMix (Evrogen, Moscow, Russia), 1 µL of each primer (10 µM), 5 µL of genomic DNA (5–50 ng/µL), and nuclease-free water up to 25 µL. The PCR was performed in an Applied Biosystems thermal cycler under the following cycling conditions:

Initial denaturation at 95 °C for 5 min;

A total of 35 cycles of denaturation at 95 °C for 30 s, annealing at 48 °C for 45 s, and elongation at 72 °C for 30 s;

Final extension at 72 °C for 7 min [36].

PCR Detection of *Ch*. *trachomatis*.

The primers NRO (forward: 5′-CTCAACTGTAACTGCGTATTT-3′) and NLO (reverse: 5′-ATGAAAAAACTCTTGAAATCG-3′) were used to amplify a 1087 bp fragment of *C*. *trachomatis* DNA.

The PCR mix included 5 µL of 5× ScreenMix (Evrogen, Moscow, Russia), 1 µL of each primer (10 µM), 5 µL of genomic DNA, and nuclease-free water to a final volume of 25 µL. The amplification protocol included the following:

Initial denaturation at 95 °C for 4 min;

A total of 49 cycles of denaturation at 95 °C for 1 min, annealing at 55 °C for 1 min, and elongation at 72 °C for 1.5 min;

Final extension at 72 °C for 7 min [37].

PCR Detection of *Brucella* spp.

For the detection of *Brucella* spp., the same primers as those used for *P*. *multocida* were used: KMT1T7 (forward: ATC CGC TAT TTA CCC AGT GG) and KMT1SP6 (reverse: GCT GTA AAC GAA CTC GC AC), producing a 527 bp amplicon.

The PCR was performed in a 25 µL reaction containing 2.5 µL of 10× buffer, 1 µL of each primer, 0.2 µL of Assu Prime Taq polymerase, 3 µL of DNA template, and 17.3 µL of nuclease-free water. The cycling conditions were as follows:

Initial denaturation at 94 °C for 5 min;

A total of 40 cycles of denaturation at 94 °C for 30 s, annealing at 55 °C for 30 s, and elongation at 68 °C for 1 min;

Final extension at 68 °C for 10 min [38].

Detection of Parasitic Pathogens *T*. *canis* and *E*. *granulosus* in Dog Feces Using PCR.

A total of 102 fecal samples from stray dogs were examined for parasitic diseases using the polymerase chain reaction (PCR) method. The analysis was performed with species-specific primers targeting the DNA of pathogenic helminths, including the nematode *T*. *canis* and the cestode *E*. *granulosus*.

PCR Detection of *T*. *canis*.

For the detection of *T*. *canis*, specific primers were used: the forward primer Tcan1 (5′-AGTATGATGGGCGCGCCAAT-3′) and the reverse primer NC2 (5′-TAGTTTCTTTTCCTCCGCT-3′), generating a PCR product of 380 bp. The reaction was carried out using the ScreenMix kit (Evrogen, Moscow, Russia). The reaction mixture contained 5 µL of 5× ScreenMix, 1 µL each of the forward and reverse primers (10 µM), 5 µL of genomic DNA (concentration ranging from 5 to 50 ng/µL), and nuclease-free water to a final volume of 25 µL. Amplification was conducted using a thermal cycler (Applied Biosystems, Foster City, CA, USA). The PCR protocol included an initial denaturation at 94 °C for 30 s, followed by 35 cycles of denaturation at 94 °C for 60 s, annealing at 58 °C for 30 s, and extension at 72 °C for 30 s. A final extension was performed at 72 °C for 10 min [39].

PCR Detection of *E*. *granulosus*.

The detection of *E*. *granulosus* was performed using the following primers: the forward primer Eg1121a (5′-GAATGCAAGCAGCAGATG-3′) and the reverse primer Eg1122a (5′-GAGATGAGTGAGAAGGAGTG-3′), yielding a PCR product of 133 bp. The PCR assay was carried out using the ScreenMix kit (Evrogen, Moscow, Russia). The reaction mixture consisted of 5 µL of 5× ScreenMix, 1 µL each of the forward and reverse primers (10 µM), 5 µL of genomic DNA (5–50 ng/µL), and nuclease-free water to reach a final volume of 25 µL. Amplification was conducted using a thermal cycler (Applied Biosystems, Foster City, CA, USA). The cycling conditions were as follows: initial denaturation at 94 °C for 5 min; 35 cycles of denaturation at 94 °C for 30 s, annealing at 55 °C for 1 min, and extension at 72 °C for 1 min; followed by a final extension at 72 °C for 5 min [40].

Electrophoretic Analysis of PCR Products.

To analyze the PCR products, 5 µL of each amplicon was mixed with DNA loading dye and subjected to electrophoresis in a 1.5% agarose gel containing 0.5 µg/mL ethidium bromide (EtBr) in 1× Tris–acetate–EDTA (TAE) buffer at 120 V for 20 min. The resulting bands were visualized and documented using a gel documentation system (Bio-Rad, Hercules, CA, USA) [41].

Analysis of Data Provided by the Veterinary Services of Uralsk and the West Kazakhstan Region.

The analysis incorporated data provided by the Veterinary Department of the West Kazakhstan Region and the Uralsk City Territorial Inspection for Veterinary Control and Supervision for the period from 2020 to 2024.

## 3. Results

### 3.1. Results of Blood and Urine Testing for Infectious Diseases Using PCR

Polymerase chain reaction (PCR) was employed to detect infectious agents in blood and urine samples from stray dogs. The PCR amplification of the blood samples yielded products of approximately 1500 base pairs, indicating the presence of bacterial pathogen DNA in these specimens. To validate these findings, urine samples were also analyzed using PCR with pathogen-specific primers to detect bacterial DNA. The presence of bacterial DNA in the urine samples confirmed ongoing infectious processes. For further identification of the detected microorganisms, positive samples were analyzed using PCR with species-specific primers targeting the DNA of the causative agents of leptospirosis (*Leptospira* spp.), pasteurellosis (*P*. *multocida*), brucellosis (*Brucella* spp.), and chlamydiosis (*Chlamydia* spp.). According to the PCR results, no DNA of *Leptospira* spp., *P*. *multocida*, or *Chlamydia* spp. was detected in either blood or urine samples. However, the DNA of Brucella spp. was identified in five samples. To further identify the *Brucella* species, AMOS-PCR (*Abortus-Melitensis-Ovis-Suis* PCR) was performed, enabling differentiation among *Brucella abortus* (Buchanan, 1914), *Brucella melitensis* (Hughes, 1893), *Brucella ovis* (Buddle and Boyes, 1953), *Brucella suis* (Tulloch, 1925), and *B*. *canis*. The PCR amplification showed no presence of DNA from *B*. *abortus*, *B*. *melitensis*, *B*. *ovis*, or *B*. *suis* in the analyzed blood and urine samples. In contrast, the DNA of *B*. *canis* was detected in five samples (Table 2).

### 3.2. Results of ELISA Testing of Canine Serum Samples for Infectious Diseases

The results of the serological testing of canine blood serum samples for infectious diseases using the ELISA are presented below for each pathogen (see Figure 2):

*Pasteurella multocida*: Of the 102 samples tested, 21 (20.6%) showed a positive reaction. This indicates a relatively high level of infection with this microorganism, which can cause respiratory diseases, as well as skin and soft tissue lesions.

*Leptospira* spp.: A positive result was obtained in only three cases, corresponding to 2.9%. This low percentage may indicate the limited spread of leptospirosis within the studied population.

*Brucella* spp.: The highest number of positive results was observed for Brucella spp., with 59 out of 102 samples testing positive (57.8%). This suggests a potentially high epidemiological significance of brucellosis in the dog population, especially considering the zoonotic nature of the disease.

*Listeria monocytogenes*: Eight samples tested positive, accounting for 7.8%. This level of infection may be attributed to the presence of the bacteria in the environment or food, posing a potential risk to both animals and humans.

*Mycobacterium* spp.: Nineteen positive reactions were observed, representing 18.6% of the samples. This result suggests the presence of mycobacterial infections in the dog population, potentially associated with atypical mycobacteria or tuberculosis pathogens.

*Chlamydia trachomatis*: No positive results were detected, indicating a 0% prevalence. This may suggest either the absence of the circulation of this pathogen among the examined animals or its minimal epidemiological role within the studied population.

### 3.3. Results of Parasitological Examination of Fecal Samples from Stray Dogs Using the Fülleborn Flotation Method

This study identified the eggs of six helminth species in the feces of stray dogs. Taxonomically, the findings included one species from the class Trematoda—*O. felineus*, two species from the class Cestoda—*Taeniidae* (Luhe, 1910) and *D. caninum*, and three species from the class Nematoda—*Toxascaris leonine* (von Linstow, 1902), *T. canis*, and *A. caninum*. Microscopic Examination of Fecal Samples from Stray Dogs Using the Fülleborn Method (Table 3).

The results of this study indicate the widespread prevalence of parasitic diseases in stray dogs. The highest extent of infection was noted for *A*. *caninum* and *D*. *caninum*, which suggests a high parasitic burden and an unfavorable sanitary and epidemiological situation. Significant prevalence was also observed for *T*. *canis* and *T*. *leonina*, which pose a potential threat to human health. To confirm the results of the helminthological studies, a PCR analysis of the dog feces was conducted to detect the most pathogenic helminth species: *E*. *granulosus* (class Cestoda) and *T*. *canis* (class Nematoda).

### 3.4. Results of Dog Fecal Sample Analysis for Parasitic Diseases Using PCR Method

The molecular analysis revealed the prevalence of the two most pathogenic helminth species: *T*. *canis* (class Nematoda) and *E*. *granulosus* (class Cestoda). The DNA of *T*. *canis* was found in 40 out of 102 samples, representing 39.2% of the total number of samples analyzed. These data indicate a high level of infection in the population of stray dogs with this parasitic species, highlighting the epidemiological significance of toxocariasis as a zoonotic disease.

The DNA of *E*. *granulosus* was detected in 17 samples, corresponding to 16.6% of the total samples. Although the prevalence is lower than that of toxocariasis, echinococcosis poses a serious threat due to the high pathogenicity of the causative agent and the risk of transmission to humans, as well as to domestic and livestock animals. In the positive samples, the DNA of the respective pathogens was amplified: PCR products of 380 base pairs for *T*. *canis* and 133 base pairs for *E*. *granulosus*, which reliably confirms their presence in the investigated material. The results of the comprehensive investigation of fecal samples from stray dogs, conducted using both the traditional Fülleborn method and molecular PCR analysis, indicate a high degree of helminth infection across various taxa in the animals (Table 4).

The Fülleborn flotation method revealed a wide spectrum of helminthiases, identifying eggs of six helminth species belonging to the classes Trematoda, Cestoda, and Nematoda. The additional use of PCR analysis confirmed and significantly complemented the microscopic findings. This highly sensitive molecular technique enabled the identification of *T*. *canis* and *E*. *granulosus* DNA—helminths of major zoonotic importance. *T*. *canis* was detected in 39.2% of the examined dogs, which is consistent with the Fülleborn method results but provides a more accurate confirmation at the molecular level. *E*. *granulosus*, which was not detected microscopically, was identified in 16.6% of the dogs exclusively via PCR, underscoring the indispensability of this method for detecting low-egg-shedding parasites or latent infections.

### 3.5. Results of Serum Analysis in Dogs for Parasitic Diseases Using ELISA

The results of the antibody analysis for *T*. *canis* revealed that 52 out of 102 dogs (50.9%) tested seropositive. This indicates a relatively high level of infection with this nematode, which is common among carnivorous animals and represents a potential zoonotic risk, particularly to children.

Even more significant findings were obtained in the examination for *E*. *granulosus*. In this case, 78 samples out of 102 tested positive, corresponding to 76.4%. Such a high level of seropositivity indicates a pronounced epidemiological concern regarding echinococcosis among dogs. Given the zoonotic nature of this disease and its severe consequences for humans, these results are of serious concern and call for veterinary–sanitary and epidemiological control measures (Figure 3).

### 3.6. Analysis of Data Provided by the Veterinary Service of Uralsk and the West Kazakhstan Region

In our analysis, we also used data kindly provided by the Veterinary Department of the West Kazakhstan Region and the Uralsk City Territorial Inspection for Veterinary Control and Supervision for the period from 2020 to 2024. The data showed that a total of 7368 dogs in Uralsk were registered in the electronic information system, microchipped, and included in veterinary check-ups. However, a steady increase in the number of stray dogs was observed during the same period. Specifically, the numbers of recorded stray dogs were 1214 in 2020, 1033 in 2021, 1765 in 2022, 1645 in 2023, and 1711 in 2024. During the study period, two cases of rabies were documented among stray dogs, along with a high prevalence of helminth infections. In total, eight helminth species from three classes—Trematoda, Cestoda, and Nematoda—were identified: Class Trematoda: *O*. *felineus*—mean prevalence (EI): 29.9%. Class Cestoda: *E*. *granulosus*—mean prevalence: 15.0%; *D*. *caninum*—mean prevalence: 54.9%. Class Nematoda: *T*. *leonina*—mean prevalence: 69.9%; *T*. *canis*—mean prevalence: 72.0%; *A*. *caninum*—mean prevalence: 75.0%; *Uncinaria stenocephala* (Railliet, 1884)—mean prevalence: 100.0%; *Dirofilaria repens*—mean prevalence: 29.4% (Table 5).

According to the Uralsk City Department of Sanitary and Epidemiological Control, several zoonotic infections were reported in the human population between 2020 and 2024. These included Brucellosis—an average of 2.2 cases per year, Echinococcosis—13.2 cases/year, Microsporia—9.4 cases/year, and Trichophytia—0.4 cases/year. The results of this analysis suggest that the increasing number of stray dogs and their high infection rates with parasitic and infectious agents have a direct impact on the city’s epidemiological situation, posing additional risks to public health. These findings highlight the ongoing circulation of zoonotic and zooanthroponotic diseases among both animals and humans in Uralsk.

## 4. Discussion

These findings align with our earlier research focused on evaluating the epizootic status of these diseases in the canine population of Uralsk [11,13,14,15,42].

The findings obtained through molecular and serological analyses indicate a high prevalence of bacterial infectious diseases among free-roaming dogs in the urban area of Uralsk. The detection of *B*. *canis* DNA in blood and urine samples by PCR, along with the presence of antibodies against *Brucella* spp., *P*. *multocida*, *Mycobacterium* spp., *L*. *monocytogenes*, and *Leptospira* spp. in serum samples, confirms the circulation of these pathogens within the stray dog population [5,7].

Of particular concern is the high seropositivity for brucellosis—57.8%. This rate is consistent with results reported in other regions, where the high prevalence of *B*. *canis* has also been documented among dogs. For example, studies from shelters in Novosibirsk and the surrounding region revealed positive reactions (antibody titers ≥ 1:200) in 50% and 58.3% of dogs, respectively [43]. Canine brucellosis represents a significant zoonotic threat. According to Menshenina V.S., in neighboring countries, this disease receives insufficient attention despite high detection rates ranging from 16.6% to 72.5%, which may pose a serious risk to public health [44].

Also noteworthy is the detection of *Mycobacterium tuberculosis* (Zopf, 1883) and *Mycobacterium bovis* (Bergey et al., 1923) *DNA* in dogs, as reported by Kalmykov V.M., Naimanov A.Kh., and Kalmykova M.S. in studies conducted in the Republic of Kalmykia. These findings underscore the potential for dogs to become infected with mycobacteria pathogenic to humans and highlight the necessity of ongoing surveillance for such pathogens in stray animal populations [45]. Collectively, the data generated via molecular and serological methods emphasize the significant epizootiological and epidemiological relevance of stray dogs as potential reservoirs of zoonotic pathogens. Considering the confirmed presence of *B*. *canis*, *Mycobacterium* spp., *Leptospira* spp., and other agents, the implementation of robust disease control and prevention programs targeting unregistered animals is imperative.

In addition to infectious diseases, parasitic infections also pose a serious threat to both animal and human health, as evidenced by their high prevalence among free-roaming dogs.

The results of our comprehensive investigation demonstrate a considerable burden of parasitic diseases among stray dogs in the urban area of Uralsk. Application of the Fülleborn flotation method, PCR diagnostics, and the ELISA enabled the detection of a wide range of helminth infections with significant epizootiological and epidemiological implications. Among the traditional diagnostic methods, the Fülleborn method proved to be the most informative, revealing six helminth species, including several of zoonotic concern: *A*. *caninum* (prevalence—35.3%), *T*. *canis* (32.3%), *D*. *caninum* (34.3%), and *O*. *felineus* (29.6%). The particularly high prevalence of *A*. *caninum* indicates a severe epizootiological situation and a significant risk of environmental contamination with infective stages. The detection of *O*. *felineus* in 29.6% of dogs emphasizes the role of stray animals in maintaining natural foci of opisthorchiasis, especially in areas with endemic water sources and food practices [5,10,11,13].

The PCR diagnostics demonstrated high sensitivity, particularly in detecting latent infections. For instance, *T*. *canis* DNA was amplified in 39.2% of the samples, while *E*. *granulosus* DNA was identified in 16.6%. Notably, *E*. *granulosus* was not detected by microscopy, likely due to its low egg production and the latent nature of some infections, highlighting the critical importance of PCR in monitoring parasitic threats, particularly those of zoonotic origin [8,9].

Serological investigations using the ELISA revealed even higher prevalence rates: 50.9% of the dogs had antibodies against *T*. *canis*, and 76.4% tested positive for *E*. *granulosus*, indicating the active circulation of these pathogens and frequent exposure among the dog population. High seroprevalence may reflect both current and past infections, making the ELISA a valuable tool for assessing the overall epizootiological tension.

A comparison with other regional studies supports the global relevance of the findings. For example, Shalmenov M.Sh. and Yastreb V.B. (2024) reported that 15.9% of dogs in Western Kazakhstan were infected with *E*. *granulosus*, consistent with our PCR (16.6%) and ELISA (76.4%) results, reaffirming the importance of echinococcosis in areas with intensive livestock farming and large stray dog populations [46]. The international literature also highlights the epidemiological role of domestic and stray animals in the transmission of helminthiases. In a study conducted in Thailand, the prevalence of *Opisthorchis viverrini* (Poirer, 1886) was 30.9% in cats and 10.2% in dogs [30]. In our study, the prevalence of *O*. *felineus* in dogs reached 29.6%, which may be attributed to the geographic and ecological characteristics of the Uralsk region, as well as local feeding practices [11,14,15,47]. The integration of multiple diagnostic approaches enabled a stratified assessment of the parasitic situation. For *T*. *canis*, the Fülleborn method revealed a prevalence of 32.3%, while the ELISA detected seropositivity in 50.9%, and the PCR confirmed the presence of parasite DNA in 39.2% of the samples. This is consistent with the concept that different diagnostic tools detect different stages of infection: PCR identifies active infections, the ELISA captures both current and past exposures, and microscopy detects egg shedding into the environment. In summary, the high prevalence of helminth infections among stray dogs necessitates a comprehensive approach to veterinary–sanitary control, including regular deworming, the establishment of a monitoring system, population control measures, and increased public awareness regarding zoonotic risks.

From a One Health perspective, the findings underscore the complex interplay between environmental health, human well-being, and the condition of free-roaming animals [48,49]. The high prevalence of both bacterial and parasitic zoonotic pathogens among stray dogs in Uralsk likely reflects broader ecological and socio-sanitary challenges, including uncontrolled waste disposal, inadequate access to veterinary care, and favorable environmental conditions for pathogen persistence [50]. Stray dogs often scavenge in landfills and interact with wildlife and livestock, acting as epidemiological bridges between ecosystems. This scenario raises concerns about environmental contamination, the silent spread of zoonoses, and the potential exposure of humans, particularly children and agricultural workers, to infectious agents. The observed infection patterns, such as the widespread presence of *T*. *canis*, *E*. *granulosus*, and *B*. *canis*, may indicate chronic health stress in the animals due to malnutrition, co-infections, and harsh environmental conditions. Addressing these issues requires an integrated surveillance strategy and intersectoral collaboration between the veterinary, medical, and environmental health sectors, in line with the One Health framework.

## 5. Conclusions

Molecular, serological, and parasitological studies have shown that stray dogs in Uralsk (West Kazakhstan) serve as an active reservoir of bacterial and parasitic zooanthroponoses. The detection of bacterial and parasitic infections indicates a high epizootiological risk.

The circulation of hazardous infectious agents within the urban fauna reflects a tense sanitary and epidemiological situation and underscores the need for a systematic approach to its stabilization. Key measures include the establishment and implementation of a monitoring system for infectious and parasitic diseases, as well as raising public awareness and improving sanitary practices.

The implementation of a comprehensive preventive framework is essential for reducing infection risks in both animals and humans, thereby ensuring sanitary and epizootic safety in the region.

The findings align with United Nations Sustainable Development Goal (SDG) 3: “Good Health and Well-Being”, as the spread of infectious and parasitic diseases among dogs in urban settings poses a significant threat to public health. These results may serve as a foundation for the development of effective strategies for the prevention and control of zoonotic infections aimed at mitigating health risks to the population [29].

## Figures and Tables

**Figure 1 biology-14-00683-f001:**
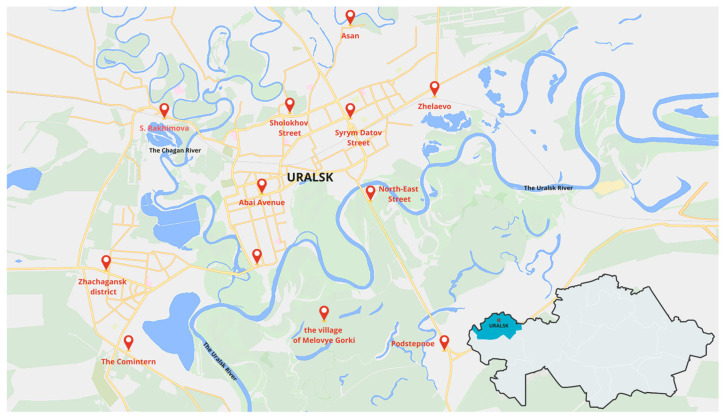
Map of the location of the city of Uralsk and stray-dog-catching points (marked in red). The insert in the lower right corner shows the Republic of Kazakhstan, where the West Kazakhstan region is highlighted in turquoise and the city of Uralsk is a dot in burgundy.

**Figure 2 biology-14-00683-f002:**
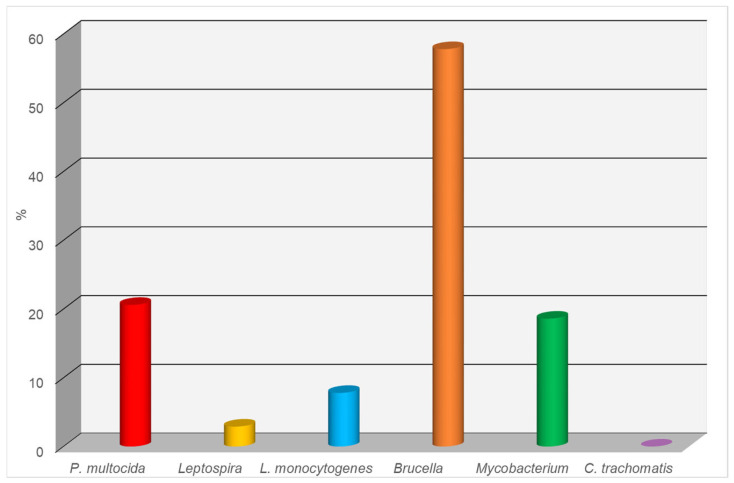
Seroprevalence of infectious agents in dogs (ELISA-based detection).

**Figure 3 biology-14-00683-f003:**
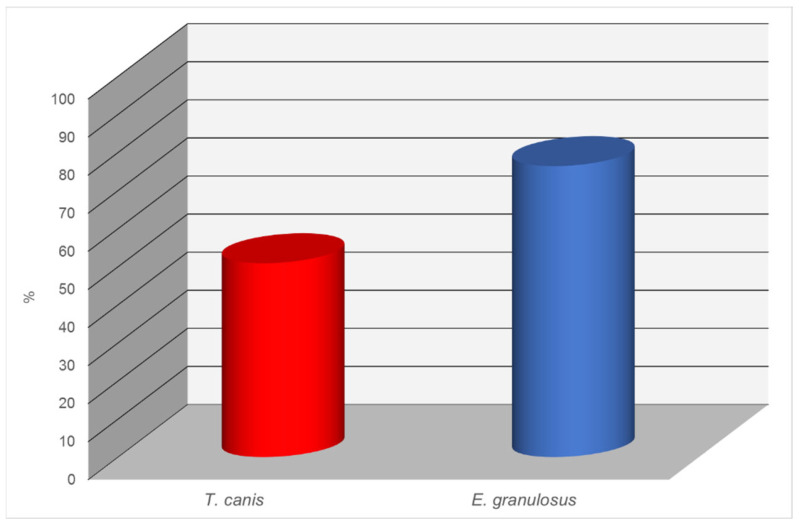
Seroprevalence of parasitic infections in dogs detected by ELISA.

**Table 1 biology-14-00683-t001:** Detection of infectious and parasitic pathogens in dogs.

No.	Pathogen	Method of Investigation	Biological Samples
Infectious Diseases			
1	*Pasteurella multocida*	PCR, ELISA	Blood, urine, serum
2	*Leptospira* spp.	PCR, ELISA	Blood, urine, serum
3	*Chlamydia trachomatis*	PCR, ELISA	Blood, urine, serum
4	*Brucella* spp.	PCR, ELISA	Blood, urine, serum
5	*Listeria monocytogenes*	ELISA	Serum
6	*Mycobacterium* spp.	ELISA	Serum
Parasitic Diseases			
7	*Toxocara canis*	PCR, ELISA	Feces, blood serum
8	*Echinococcus granulosus*	PCR, ELISA	Feces, blood serum

**Table 2 biology-14-00683-t002:** PCR detection of infectious diseases in blood samples of stray dogs.

Disease Name	Number of Blood and Urine Samples Tested	Number of Positive Blood Samples (n/%)	Number of Positive Urine Samples (n/%)
Leptospirosis	102	0	0
Pasteurellosis	102	0	0
Brucellosis	102	5/4.9	5/4.9
Chlamydiosis	102	0	0

**Table 3 biology-14-00683-t003:** Prevalence of intestinal helminths in stray dogs in Uralsk.

Helminth Species	Animals Examined (n)	Animals Infected (n)	EI (%)	II Eggs/Animal
*Opisthorchis felineus*	102	32	29.6	18.4
*Family taeniidae*	102	16	14.8	15.6
*Dipylidium caninum*	102	35	34.3	9.6
*Toxascaris leonina*	102	30	29.4	18.5
*Toxocara canis*	102	33	32.3	14.2
*Ancylostoma caninum*	102	36	35.3	23.6

**Table 4 biology-14-00683-t004:** Results of dog fecal samples for the detection of parasitic diseases using the PCR method.

Parasites	Number of Examined Specimens	Number of Positive Cases	%
*Toxocara canis*	102	40	39.2
*Echinococcus granulosus*	102	17	16.6

**Table 5 biology-14-00683-t005:** Prevalence of helminth infections in dogs according to the data of the veterinary service of Uralsk city and the West Kazakhstan region for 2020–2024.

No.	Helminth Species	Examined (n)	Infected (n)	Infection Rate, %
	Class Trematoda			
1	*Opisthorchis felineus*	7368	2203	29.9
	Class Cestoda			
2	*Echinococcus granulosus*	7368	1105	15.0
3	*Dipylidium caninum*	7368	4045	54.9
	Class Nematoda			
4	*Toxascaris leonina*	7368	5150	69.9
5	*Toxocara canis*	7368	5305	72.0
6	*Ancylostoma caninum*	7368	5526	75.0
7	*Uncinaria stenocephala*	7368	7368	100.0
8	*Dirofilaria repens*	7368	2166	29.4

## Data Availability

This study is based on newly generated diagnostic data from stray dog populations in the Urals metropolitan area. Due to ethical restrictions and the nature of the data, only summarized results are presented in the article. Detailed data are available from the corresponding author upon reasonable request.

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
