# Peer review of "Stray Dogs as Reservoirs and Sources of Infectious and Parasitic Diseases in the Environment of the City of Uralsk in Western Kazakhstan"

_biology, 2025, doi:10.3390/biology14060683_

Round 1
Reviewer 1 Report
Comments and Suggestions for Authors
The manuscript is well-structured and presents a representative sampling to assess the health status of dogs in the city. I have a few comments:
Include the name of the country in the title of the manuscript.
Include a map of the city where the samples were collected, highlighting it within the state and country maps. Were GPS coordinates used?
Clarify the situation of the animals included in the sample. Were they stray, semi-domesticated, or domesticated animals?
Were all dogs tested for all variables?
What was the sampling criterion used for selecting the animals?
Discuss environmental health issues and relate them to the results found, attempting to hypothesize the animals' health status from a One Health perspective.
Author Response
Response to Reviewer 1 Comments
|
||
1. Summary |
|
|
Thank you very much for taking the time to review this manuscript. Please find the detailed responses below and the corresponding revisions/corrections highlighted/in track changes in the re-submitted files. |
||
2. Questions for General Evaluation |
Reviewer’s Evaluation |
Response and Revisions |
Does the introduction provide sufficient background and include all relevant references? |
Yes |
Thank you. The reviewer’s positive evaluation is appreciated. |
Are all the cited references relevant to the research? |
Yes |
Thank you for your positive feedback on the research design. |
Is the research design appropriate? |
Yes |
We are grateful for your evaluation and are glad the methods were found to be appropriate. |
Are the methods adequately described? |
Yes |
Thank you. We appreciate the confirmation that the results are clearly presented. |
Are the results clearly presented? |
Yes |
Thank you for your kind assessment. |
Are the conclusions supported by the results? |
- |
This section was left blank by the reviewer. We understand that no changes are required regarding the figures and tables. However, we remain open to making any revisions if needed. |
3. Point-by-point response to Comments and Suggestions for Authors |
||
Comments 1: Include the name of the country in the title of the manuscript.
|
||
Response 1: We agree that including the country name will improve the clarity and specificity of the title. Therefore, we have revised the title to include the country name. The new title is: "Stray Dogs as Reservoirs and Sources of Infectious and Parasitic Diseases in the Urban Environment of the Ural Metropolis, Western Kazakhstan" (lines 1–4, page 1).
|
||
Comments 2: Include a map of the city where the samples were collected, highlighting it within the state and country maps. Were GPS coordinates used? |
||
Response 2: We agree with the reviewer’s comment. Accordingly, we have included a map of the Republic of Kazakhstan and a detailed map of Uralsk city in the revised manuscript. The map highlights the sample collection sites (dog capture points), which are indicated with blue flags. Although GPS coordinates were not used during sampling, we have described the method for determining the capture locations based on administrative data and field records. This information has been added as Figure 1 – “Map showing the location of Uralsk city, West Kazakhstan Region, Republic of Kazakhstan, blue flags – stray dog capture points.” This figure and explanation are provided on page 193, line 1 of the revised manuscript.
Comments 3: Clarify the situation of the animals included in the sample. Were they stray, semi-domesticated, or domesticated animals? Response 3: We appreciate the reviewer’s suggestion. We have clarified the definition and status of the animals included in the study. Specifically, we have provided an explanation regarding the classification of stray dogs, distinguishing them from semi-domesticated and domesticated animals. This clarification has been added to the revised manuscript on page [4], lines 161–163.
Comments 4: Were all dogs tested for all variables? Response 4: All dogs were tested for all variables according to the applied diagnostic methods.
Comments 5: What was the sampling criterion used for selecting the animals? Response 5: Thank you for the valuable comment. We have added a clarification to the manuscript regarding whether all dogs were tested for all variables. This explanation can be found in the revised version on lines 183–185. Comments 6: Discuss environmental health issues and relate them to the results found, attempting to hypothesize the animals' health status from a One Health perspective. Response 6: Thank you for this insightful comment. In response, we have added a discussion addressing environmental health issues and their relationship to the results obtained in our study. This section also provides a hypothesis regarding the animals' health status from a One Health perspective. The added content can be found in lines 655–669 and 683–687 of the revised manuscript. |
||
4. Response to Comments on the Quality of English Language |
||
Point 1: The English is fine and does not require any improvement |
||
Response 1: Thank you. We appreciate the positive feedback regarding the quality of the English language. No changes were required in this regard. |
Reviewer 2 Report
Comments and Suggestions for Authors
The manuscript “Epidemiological Role of Infectious and Parasitic Diseases in Stray Dogs in the Context of an Urbanized Metropolis: A Case Study from Uralsk” by Nametov et al. presents interesting findings on infectious and parasitic agents from stray dogs in Kazakhstan. While the topic is of particular interest in both the One Health and Public Health frameworks, I believe the manuscript will require much more work for it to be published in a scientific journal.
I will break down some of the major flaws that in my opinion need to be addressed before resubmitting the manuscript again:
- All scientific names of the pathogens must be formatted in Italics throughout the whole manuscript!
Simple summary:
- For my personal preference I’d like papers to be written in an impersonal form, for instance “We caught over one thousand stray dogs” (Line 20) could be changed into “Over one thousand stray dogs were captured”. That’s just a personal preference, so if the editor is fine with it you can also leave it as is.
Abstract:
- The results section of the abstract (Lines 37-42) is hard to read and should be rephrased. I would give first all the serology results (data on the seroprevalence of Pasteurella multocida, Mycobacterium spp., Listeria monocytogenes, and Leptospira spp. are missing), and then give all the molecular results.
Introduction:
- The introduction needs much more referencing, the paragraphs are sustained by just one (or two in one case) references, a much thorough literature review should be carried out to better delineate the topic of the article.
- The first three paragraphs (Lines 49-60), as well as the second-to-last one (Lines 108-114), completely lack supporting literature. As said in the previous point, more references should be added to back up the authors’ claims.
Materials and Methods:
- This section is rather chaotic with that extreme sectioning and I believe it should be restructured to allow for better understanding of your laboratory procedures. For instance, the collection of samples is given in sections 2.6.1-2.6.4. Sample collection should be addressed before talking about the laboratory procedures carried out on the samples collected.
- I would divide the M&M in just 4 sections: Sample collection and Clinical Examination; Coprological pathogen detection; Serological pathogen detection; Molecular pathogen detection.
- Line 135: describe the capturing methods to demonstrate they align with animal welfare policies.
- Line 137: provide the accession link to the website and the date you accessed the website
- Line 143: you could provide the database as supporting information
- Line 146: I believe the term “molecular-genetic” is redundant, just go with “molecular”
-Lines 150-153: why describing the PCR reagents before giving the DNA extraction details?
-Lines 156-206: I am assuming the primers and thermal profiles were not designed ex novo for this study, is that the case? If not (i.e., the primers and thermal profiles were sourced from literature) please add appropriate referencing where needed
-Line 212: the word “titled” is not appropriate here
-Line 225-226: Those are results, remove the sentence from “In the presence…”
-Line 230: Insert the specifics of the microplate reader
- Lines 238-258: Lines 156-206: I am assuming the primers and thermal profiles were not designed ex novo for this study, is that the case? If not please add appropriate referencing where needed
Results:
-Lines 340-341 and 357-358: the names of the disease should not be formatted in italics
-Lines 368-388: section 3.2.2. reports exactly the same results reported in the previous section. Table 2 and Table 3 are exactly identical…
- Figure 1 is hard to read, the labels on the x axis are too small to be read. I suggest reducing the scale on the y axis so that the bars will be bigger an easier to interpretate
- Lines 417-419: report once again the result of the PCR on bacterial pathogens, one time is enough
- Section 3.2.4.: Cestoda, Nematoda, Taeniidae, Biohelminths and Geohelminths should not be formatted in italics, furthermore Taeniidae is not a genus name hence it should not be followed by “spp.”
- Figure 2 is hard to read, the labels on the x axis are too small to be read. I suggest reducing the scale on the y axis so that the bars will be bigger an easier to interpretate
- Table 4: substitute “Nosological Entity” with “the Parasite”
- Line 500-501: is true that PCR is very sensitive but it should also be taken in account that some parasites like cestodes sheds their eggs with intermittent patterns, therefore an animal can be at the same time harbor the parasite but result as negative to PCR parasite detection
- Line 530-531: the figure between those lines is named “Figure 2” but there is already a Figure 2 in the manuscript
- It would be interesting to add the titres of the samples that tested positive to serology
Discussion:
- As in the introduction section the Discussion as well lacks of referencing. I advise a more thorough analysis of the existing literature to compare to the results obtained
-Conclusions:
-Conclusions are too long, it should be a brief paragraph summing up the paper and possibly opening to future perspectives.
All things considered I advise the editor to reject this manuscript for publication in this form but I encourage the authors to thoroughly revise and correct the manuscript before resubmitting it again.
Author Response
Please see the attachment." in the box if you only upload an attachment

Reviewer 3 Report
Comments and Suggestions for Authors
Stray dogs may harbor a range of viruses, bacteria and parasites with potential zoonotic risk, which poses a public problem in many countries. The problem studied by the Authors of the article are very important from the point of view of the spreading of zoonotic infections between stray dogs, domestic animals and humans. Therefore, research Kazakhstan colleagues is always relevant. I congratulate the authors for the extensive effort made in carrying out a good scientific paper.
But I have some remarks about this manuscript, which will undoubtedly improve the article.
Not everyone knows where Uralsk is, so the title of the article should be added “… from Uralsk, Western Kazakhstan”
What about Dirofilaria species (Onchocercidae) in stray dogs that are also known as causative agents of zoonotic helminthiasis of dogs and human? Must be specified in the Introduction.
Lines 86,328,466, etc. – The term “infestation” is more appropriate for ectoparasites. For helminths, infection is more suitable.
The aim of the work is somewhat different. In the paper the Authors did not consider the role of dogs in transmission, but considered the role of stray dogs in the spread and preservation of infectious and parasitic diseases. This needs to be corrected.
Lines 122-128 - Information about bioethics should be placed in the appropriate section at the end of the article. It should be removed here.
Lines 129-133 – And this information is good for Conclusion. Please, rearrange the text.
Lines 441-445 - This refers to Materials and Methods. It should be moved to this section. And Line 442 – “Biohelminths” - This term is known only to helminthologists of the former USSR. The whole world does not know this term. And they use the phrase “helminths with an indirect life cycle.“Geohelminths” you can also use terms “helminths with a direct life cycle or “Soil Transmitted Helminths” (STH)
It is better to present the results of the study in the form of a table, without overloading the text with numerical data. This is especially true for section 3.2.4.
Authors should separate Results and Discussion. Results should contain information about the results obtained in the work. But, in Results the authors immediately discuss the data obtained (3.1, 3.2.3, 3.2.4 and 3.2.5 sections) and even provide references to other works (lines 564), which is not welcome. Please move this to Discussion.
The same applies to the Conclusions for each section of Results. They should be moved to the Discussion/Conclusion. Like in Lines 508-512, 528-538, 556-579, etc. Otherwise your Discussion is good but short. This needs to be corrected.
The Сonclusion needs to be restructured. Here you need to clearly and concisely present your findings and recommendations without numerical data and subheadings.
The Materials and Methods do not contain a methodology for studying dog feces. However, the Results contain it (Lines 431-436). This should be moved to the Material and Methods section.
Lines 514-517, 476-479 – These also applies to Materials and Methods.
As far as I understand, section 3.1. is not the authors' data. Then this should, firstly, be indicated in the Materials and Methods: " In our analysis the data kindly provided by the Veterinary Department of the West Kazakhstan Region and the Uralsk City Territorial Inspection for Veterinary Control and Supervision for the period from 2020 to 2024 were also used.” And put this section at the end of the Results. Your data should go first.
According International Code of Zoological Nomenclature (ICZN) at the first mention of genus or species in article text its full Latin name with the author and year of description should be given; in relation all species. For example, in lines 68,69,70-72,75 – Toxocara canis (Werner, 1782), Toxocara cati Schrank, 1788, Trichuris vulpis Froelich, 1789, Giardia duodenalis Stiles, 1902, Ancylostoma caninum (Ercolani, 1859), Dipylidium caninum (Linnaeus, 1758), Strongyloides stercoralis (Bavay, 1876), Brucella canis Carmichael & Bruner, 1968 etc.
This must be done both at the first mention in the text and in Table 1. On subsequent mentions, the generic name is abbreviated to the first letter. Except when the sentence begins with the parasite name.
Latin names of parasites should be written in italics in the Abstract and text of the article. Please check the whole text.
Line 71 - Toxocara canis - the second mention of the species in the text of the article, therefore the generic name is abbreviated - T. canis.
Line 75 – As far as I know, Novosibirsk is in Western Siberia, and it is not the north, but in the south part of Western Siberia. It is better to write Western Siberia.
Line 592 – correct reference - According to Menshenina [?],” And by the way, in References is missing.
Line 597 - the same Kalmykov et al. [?]. In References is missing
Line 634 – Shalmenov, Yastreb [?]. Also in References is missing. Authors should carefully check the reference list.
Line 655 - replace dot with comma.
The manuscript deserves to be published, but serious corrections in text are needed.
Round 2
Reviewer 2 Report
Comments and Suggestions for Authors
Dear editor, the manuscript was greatly improved by the authors but, in my opinion, there are still things that need to be addressed before publication.
General considerations:
- In more than one instance there is no space after the period, for instance line 18 “animals.This study” instead of “animals. This study”
- “spp.” should not be formatted in italics;
- References should be put right at the end of the sentence they are backing up and not at the end of the paragraph, otherwise it will become difficult to understand which reference is relative to what concept. For instance, at lines 129-135, references should be put right after the names of the authors.
-For my personal preference there is still too much use of returns in the text that make the manuscript unnecessarily lengthy. I’d use a more compact formatting without returning after almost every sentence;
Introduction:
- The symbols in lines 79 and 80 should be substituted with “and”;
- Remove “etc.” at line 87;
- The symbols in line 120 should be substituted with “and”;
-The descriptor of D. repens should be put only the first time the name was mentioned (line 122) and not line 138.
Materials and Methods:
- Table 1, I’d remove the descriptors’ names from Table 1 as they clutter the table;
-Use the appropriate formatting for the subsection headings “Blood collection:” (line 210) and “Urine collection:” (line 223);
-“Taeniidae” should not be formatted in italics (lines 271-272);
-The reaction mix volume for the 16S rRNA gene doesn’t add up to 25 microliters (2.5+0.5+0.5+3+6=12.5), besides the volume to add of a 5x mastermix in a total volume of 25 microliters should be 5 microliters, not 2.5;
-Leptospira should be formatted in italics (line 354-355).
Results:
-Brucella should be formatted in italics (line 438);
-Figure 2 should be recalled at line 449 and not in the subsection of C. trachomatis;
-In the caption of Table 3 it says “Adult Stages” but then one of the columns reports the II of eggs detected, I would remove the specification “Adult Stages” in the caption. Again, Taeniidae should not be formatted in italics;
- Line 485 “infection” should not be formatted in bold;
-Lines 544 to 555 could be transformed in a table for more clarity;
References
- the reference list should be formatted according to the journal instructions.
Reviewer 3 Report
Comments and Suggestions for Authors
The manuscript under review deals with the study of prevalence and preservation of zoonotic parasites spreading in stray dogs from Uralsk, Western Kazakhstan. The manuscript has become much better and more logical and leaves a favorable impression.
But I still have some remarks, which are more of a technical nature:
- Authors need to put things in order with the authors of the species description. In the case when a species has several authors; one in brackets, and the other, after, without. This is not accidental. The first (in brackets) is the author (authors) who first described the species. The second (without brackets and the year of description later) is the author (authors) who transferred the species from one genus to another. Therefore, please, сheck the Latin names of species both in the tables and in the text of the manuscript. For example: Opisthorchis felineus (Rivolta, 1884) Blanchard, 1895; Chlamydia trachomatis (Busacca, 1935) Rake, 1957; Listeria monocytogenes (Murray, Webb and Swann, 1926) Pirie 1940; Toxocara canis (Werner, 1782) Stiles, 1905.
In principle, according to the rules of the ICZN, only one first author who described species is now required ...
- Latin names of parasites are given in full only the first time. The second and subsequent times the generic name is abbreviated. Please check the text, because you do not always follow this.
- Figure 1 is not quite clear and distinct. It is necessary to increase the clarity (resolution) and remake the map of Kazakhstan, leaving only the contours and the point of the city of Uralsk.
- A paragraph cannot consist of a single sentence. The Authors have many such paragraphs in the text of the manuscript. It is necessary to attach single sentences to the previous or following paragraph. For example: 78-81, 82-84, 142-143, 151-153, 239-241, 321-322, 335-340, etc.
- Lines 450,460 and further – a sentence cannot begin with an abbreviation, so in this case the generic name is written in full.
Small remarks:
The title of the publication says Ural, or should it say Uralsk?
Line s79,80 – Russian words need to be removed.
Line 101 – italics.
Line 129,130,134 – correct references in text Lai et al. [18], Tsai et al. [19], Fan [20]. Also in next paragraph.
Line 438 – Brucella italics.
Lines 476, 477 – remove extra italics.
Line 477 – Toxascaris leonina (von Linstow, 1902)
554 – First mention of the species – author and year of description needed.
546-555 - It is better to give the generic names in full here.
The manuscript can published but minor corrections in text are needed.
